# Dynamic Cerebral Autoregulation Post Endovascular Thrombectomy in Acute Ischemic Stroke

**DOI:** 10.3390/brainsci10090641

**Published:** 2020-09-16

**Authors:** Faheem Sheriff, Pedro Castro, Mariel Kozberg, Sarah LaRose, Andrew Monk, Elsa Azevedo, Karen Li, Sameen Jafari, Shyam Rao, Fadar Oliver Otite, Ayaz Khawaja, Farzaneh Sorond, Steven Feske, Can Ozan Tan, Henrikas Vaitkevicius

**Affiliations:** 1Department of Neurology, Massachusetts General Hospital, 55 Fruit Street, Boston, MA 02114, USA; mkozberg@gmail.com (M.K.); SJAFARI1@mgh.harvard.edu (S.J.); oliverotite@gmail.com (F.O.O.); AKHAWAJA@mgh.harvard.edu (A.K.); 2Department of Neurocritical Care, Brigham and Women’s Hospital, 75 Francis Street, Boston, MA 02115, USA; slmichaud@bwh.harvard.edu (S.L.); amonk@bwh.harvard.edu (A.M.); kli5@mgh.harvard.edu (K.L.); sfeske@bwh.harvard.edu (S.F.); HVAIT@bwh.harvard.edu (H.V.); 3Department of Neurology, Division of Neuroendovascular and Neurocritical Care, Texas Tech University Health Science Center, El-Paso, TX 79905, USA; 4Department of Clinical Neurosciences and Mental Health, Faculty of Medicine of University of Porto, 4200-319 Porto, Portugal; pedromacc@gmail.com (P.C.); elsaazevedo1@gmail.com (E.A.); 5Department. of Neurology, Centro Hospitalar São João, 4200-319 Porto, Portugal; 6Department of Neurocritical Care, Brown University, Providence, RI 02912, USA; shyam_rao@brown.edu; 7Department of Stroke and Neurocritical Care, Feinberg School of Medicine, Northwestern University, Chicago, IL 60611, USA; Farzaneh.Sorond@nm.org; 8Department of Physical Medicine and Rehabilitation, Spaulding Rehabilitation Hospital and Harvard Medical School, Charlestown, MA 02129, USA; cotan@mgh.harvard.edu; 9Department of Radiology, Division of Neuroradiology, Massachusetts General Hospital, 55 Fruit Street, Boston, MA 02114, USA

**Keywords:** dynamic cerebral autoregulation, endovascular thrombectomy, stroke, transcranial doppler ultrasound

## Abstract

The development of the endovascular thrombectomy (EVT) technique has revolutionized acute stroke management for patients with large vessel occlusions (LVOs). The impact of successful recanalization using an EVT on autoregulatory profiles is unknown. A more complete understanding of cerebral autoregulation in the context of EVT may assist with post-procedure hemodynamic optimization to prevent complications. We examined cerebral autoregulation in 107 patients with an LVO in the anterior circulation (proximal middle cerebral artery (M1/2) and internal cerebral artery (ICA) terminus) who had been treated using an EVT. Dynamic cerebral autoregulation was assessed at multiple time points, ranging from less than 24 h to 5 days following last seen well (LSW) time, using transcranial Doppler ultrasound recordings and transfer function analysis. Complete (Thrombolysis in Cerebral Infarction (TICI) 3) recanalization was associated with a more favorable autoregulation profile compared with TICI 2b or poorer recanalization (*p* < 0.05), which is an effect that was present after accounting for differences in the infarct volumes. Less effective autoregulation in the first 24 h following the LSW time was associated with increased rates of parenchymal hematoma types 1 and 2 hemorrhagic transformations (PH1–PH2). These data suggest that patients with incomplete recanalization and poor autoregulation (especially within the first 24 h post-LSW time) may warrant closer blood pressure monitoring and control in the first few days post ictus.

## 1. Introduction

Acute stroke management has improved dramatically over the past few years, particularly with the development of the endovascular thrombectomy (EVT) technique for large vessel occlusion (LVO) strokes [1,2,3,4,5]. Despite these advancements, a significant subset of patients do not benefit from their EVT procedure, despite successful recanalization [6]. Poor outcomes after recanalization are due to various pathophysiological mechanisms, including “reperfusion injury” (increased edema with or without hemorrhage in reperfused tissue), and the “no reflow” phenomenon, in which sufficient reperfusion of distal tissue is not achieved, despite recanalization of the proximal vessel [7].

Cerebral autoregulation is the ability of cerebral vasculature to maintain stable cerebral blood flow despite fluctuations in the systemic blood pressure through modulation of the cerebrovascular tone [8,9]. This protects the brain from ischemic and hyperemic damage over a wide range of systemic blood pressures. Cerebral autoregulation is known to be impaired post stroke, which may contribute to both a reperfusion injury and reflow failure, and may indirectly compromise outcomes. For example, post-stroke impairment in autoregulation has been associated with poor outcomes, including larger infarct volumes and increased rates of hemorrhagic conversion. These impairments are particularly prominent within the first five days post stroke [10,11], and appear to be global, affecting both the ipsilateral and the contralateral hemispheres [8,9,10,11,12,13,14,15].

However, the effect of the recanalization of affected vessels with EVT on cerebral autoregulation remains unknown. Although clinical practice varies, many stroke and critical care neurologists aim for tighter blood pressure control in patients via successful recanalization using EVT compared to those who did not undergo EVT, with the assumption that all patients will have impaired autoregulation and therefore are at a high risk of reperfusion injury after recanalization [15]. However, questions remain regarding which patients are at the highest risk of reperfusion injury, the temporal course of this risk, and the appropriate blood pressure goals. There is evidence showing that outcomes after an EVT are worse with both higher post-procedure blood pressures [16] and lower peri-procedural blood pressures [17], suggesting that cerebrovascular dysfunction may lead to either a reperfusion injury or an infarct expansion in this patient population. End-tidal carbon dioxide is also an important, potentially modifiable factor that influences cerebral blood flow, where hypocapnia is known to occur frequently in acute ischemic stroke patients [18]. This in addition to other peri-procedural factors, including the mode of anesthesia, which may have a profound impact on patient outcomes through modifying cerebral blood flow and autoregulation [19]. Thus, a more complete understanding and quantification of the autoregulatory profile of patients post EVT may help to determine which patients are at the highest risk of a reperfusion injury and/or an infarct expansion ischemia, which can be used to guide both intraoperative and post-procedural hemodynamic management. 

The objective of this study was to assess the dynamic cerebral autoregulation in patients with LVO who underwent EVT and determine the relationship between the degree of recanalization and the cerebral autoregulatory profiles over the first 5 days following their LSW time using transcranial Doppler (TCD).

## 2. Material and Methods

Adult patients with a large vessel occlusion in the anterior circulation (including the proximal middle cerebral artery (M1/2) and the internal cerebral artery (ICA) terminus), with a National Institutes of Health Stroke Scale (NIHSS) score ≥6 at any time within 72 h of the last seen well (LSW) time who underwent endovascular thrombectomy were eligible for the study. Patients were enrolled if the first TCD measurement could be performed within 120 h of the LSW time. The exclusion criteria included the inability to tolerate TCD measurements and inadequate temporal acoustic windows for TCD measurements. Patients were recruited from both the Brigham and Women’s Hospital in Boston, Massachusetts, and the Centro Hospitalar Universitário São João in Porto, Portugal, between December 2016 and January 2019. The institutional ethical standards committees at both hospitals approved this study, and written consent was obtained from each subject or healthcare proxy.

### 2.1. Data Acquisition and Analysis

Members of the stroke team found the NIHSS score at admission and discharge for each patient. The Thrombolysis in Cerebral Infarction (TICI) scores were found post thrombectomy (classified for analysis as complete (TICI 3) and incomplete (TICI 2a/2b)), with TICI 2a representing less than 50% reperfusion of the affected territory and TICI 2b representing greater than 50% reperfusion [20]. The infarct volumes were calculated using the “ABC/2”method (adapted from intracerebral hemorrhage studies [21]). Collaterals were graded using the American Society of Interventional and Therapeutic Neuroradiology/Society of Interventional Radiology (ASITN/SIR) scale modified for CT angiography CTA [22]. The hemorrhagic transformation was graded using the European Cooperative Acute Stroke Study (ECASS) criteria [23]. 

The TCD data were collected at multiple time points post-LSW time in each patient with data collected at <24 h (66 patients), 24–72 h (47 patients), 72–96 (27 patients), and/or >96 h (23 patients) after the LSW time. Bilateral 2 MHz probes were applied with a headset to insonate the bilateral proximal middle cerebral arteries (MCAs)for data acquisition. In situations in which flow velocities in the M1 segment of the MCA were not reliable due to insufficient recanalization, proximal M2 segments were insonated; if this was not possible, patients were excluded. Simultaneous blood pressure recordings were performed using either a noninvasive monitoring device (Finapres, Enschede, The Netherlands) or an arterial line if clinically indicated, as appropriate. Prior work has demonstrated similar results in dynamic cerebral autoregulation parameters using blood pressures derived from Finapres and arterial line techniques [24]. The head of the bed was maintained at 30° throughout all measurements to control for the effect of head position on cerebral autoregulation in acute ischemic stroke [25]. In the majority of patients, the end-tidal carbon dioxide (CO_2_) was continuously recorded using a nasal cannula attached to a capnograph (Nonin, Amsterdam, The Netherlands).

Transfer function analysis was used to determine the relationship between spontaneous fluctuations in the mean cerebral blood flow velocity (CBFV) and mean arterial pressure (MAP) [26]. The transfer coherence, gain, and phase shift were calculated for each subject within the standard frequency ranges (very low frequency (VLF): <0.03 Hz, and low frequency (LF): 0.03–0.07 Hz; the lower limit of the VLF range was 0.02 due to the relatively short duration of the recordings) [27]. These frequency bands were chosen based on convention [28] and on recent data suggesting that autoregulation is most effective at frequencies close to and lower than 0.03 Hz [29]. The coherence indicates the extent of the linear relationship between the arterial pressure and the cerebral blood flow fluctuations (a lower coherence indicates more effective autoregulation). The gain provides a measure of the dampening of the amplitude of the fluctuations in the blood pressure by cerebral blood vessels (a lower gain indicates more effective autoregulation). The phase shift provides a measure of the temporal relationship between the oscillations in the systemic blood pressure and the cerebral blood flow (a higher phase shift indicates more effective autoregulation). 

We have relied on spontaneous, rather than induced, blood pressure and cerebral blood flow oscillations for assessing the autoregulation. While assessing cerebral autoregulation through induced blood pressure changes has recently been shown to be a more reliable metric [30], this technique cannot be used immediately after an LVO stroke, as the induced blood pressure changes may be harmful to the patient.

### 2.2. Statistical Analysis 

We have previously shown that the coherence at the frequency bands used in this study is a relatively reliable measure of the integrity of autoregulation, although the gain and phase shift may be less reliable [29]. By definition, effective autoregulation should be a low coherence state, which in turn, creates uncertainties around the estimated gain and phase relation [31,32]. To reliably account for these uncertainties, we derived the precisions of the estimated parameters based on the confidence intervals of the estimated values for each patient and recording [33,34] and accounted for the precision using weights in our statistical analysis. However, for the ease of interpretation, all data presented throughout the text, tables, and figures are in standard units. 

Our initial analysis provided evidence of bilateral impairments in autoregulation; there were no significant hemispheric differences in the estimated transfer function (coherence, gain, and phase) between contralateral and ipsilateral sides within each frequency region and across all measurements (ANOVA *p* > 0.1 for all). Thus, the cerebral blood flow measurements of each hemisphere were used as repeated measures. The two frequency band ranges (<0.03 Hz and 0.03–0.07 Hz) were considered separately.

We used linear mixed-effects models with the group (complete (TICI 3) vs. incomplete (TICI 2a/b) recanalization) and time point (<24, 24–72, 72–96, and >96 h) as the fixed effects, and the measurement side and time point nested within each patient as random effects. This allowed us to reliably explore the differences in cerebral autoregulation while explicitly accounting for the repeated measurements and the missing data and/or time points. Whenever a significant effect was evident, we used appropriate post-hoc analyses to identify the differences across time points.

## 3. Results

A total of 136 subjects with large vessel occlusions were enrolled, of which, 107 subjects had useable data. Patients were excluded due to atrial fibrillation, a high burden of premature ventricular contractions (PVCs), and a poor signal-to-noise ratio in the recorded data. This relatively high proportion of patients with non-utilizable data was possibly reflective of the critical care population with a higher incidence of cardiac dysrhythmias and PVCs; patient agitation may have also been contributory. Of the 107 subjects, 39 showed “incomplete” recanalization (TICI 2a or 2b), and 68 showed “complete” recanalization (TICI 3) (Table 1). Of note, only one patient showed TICI 2a recanalization. Patients were age-matched across both groups, with no significant differences in the baseline stroke characteristics, tissue plasminogen activator (tPA) administration, the Alberta Stroke Program Early CT (ASPECTS) score, or collateral grade between the two groups, as assessed at the initial presentation (Table 1) [22]. Patients with complete recanalization experienced significantly higher rates of early neurological recovery than patients with incomplete recanalization, defined as an NIHSS score of 0–2 and/or an NIHSS score at discharge ≥ 4 points less than that at admission (*p* < 0.05).

Patients with incomplete (TICI 2a/b) recanalization had significantly higher infarct volumes at 24 h and lower early neurologic recovery (*p* < 0.01) compared with patients with complete (TICI 3) recanalization (Table 1). The infarct volume may have also been related to cerebral autoregulation. In fact, when dichotomized into small (<70 mL) and large (>70 mL) infarcts, which used a cut-off volume chosen based on the stroke volumes used to determine patients eligible for a thrombectomy over extended time windows (as used in the Diffusion and Perfusion Imaging Evaluation for Understanding Stroke Evolution (DEFUSE 3) study [35]), larger stroke volumes (>70 mL) were associated with a higher coherence and a lower phase on the ipsilateral side within the VLF band (time × infarct size effect *p* < 0.01 for coherence and *p* = 0.01 for gain) and LF bands (time × infarct size effect *p* = 0.01 for coherence) (also see Table 2 for the differences across all time points). To account for these differences, we used the infarct size (small vs. large) as a covariate in our statistical analyses below.

Time-dependent impairments in cerebral autoregulation were observed in both groups. Cerebral autoregulation was observed to initially worsen and gradually recover approximately 72 h after an ischemic stroke, the effects of which were significantly more pronounced in patients with incomplete recanalization (Figure 1 and Appendix A). Coherence was observed to peak in both groups 24–72 h after the stroke (*p* < 0.05 compared to the <24 h VLF-band values for both groups), and then gradually return to the <24 h values, while the gain peaked at 72–96 h (*p* < 0.05 compared to <24 h values for both the VLF and LF bands). Incomplete recanalization (TICI 2a or 2b) was associated with a higher coherence and a higher gain, i.e., worse autoregulation (significant differences were observed in the VLF band and a trend that did not achieve significance was observed in the LF band) (Figure 1; VLF coherence group effect *p* < 0.05, time effect *p* < 0.01, interaction *p* > 0.10; LF coherence group effect *p* > 0.1, time effect *p* < 0.01, interaction *p* > 0.1; VLF gain group effect *p* = 0.06, time effect *p* < 0.01, interaction *p* > 0.1; LF gain group *p* > 0.1, time effect *p* < 0.01, interaction effect *p* > 0.10; VLF phase group effect, time effect, and interaction *p* > 0.1; LF phase group and time effect *p* > 0.10, interaction *p* < 0.05). Given that the infarct volume was significantly higher in patients with incomplete recanalization and the potential effects of the infarct volume on cerebral autoregulation, this effect was accounted for in our analysis (discussed further in the Methods section). Moreover, the end-tidal CO_2_ levels tended to be higher in larger strokes (<70 vs. >70 mL, 33.3 ± 4.5 vs. 35.3 ± 4.5 mmHg, *p* = 0.06).

Impairment in the cerebral autoregulation within the first 24 h after a stroke onset was also associated with a hemorrhagic transformation (Figure 2). Among patients who had data from less than 24 h after the LSW time, coherence was increased within both the VLF and LF frequency ranges on the ipsilateral side compared to the contralateral side in those who had a parenchymal hematoma hemorrhage (PH1 and PH2), as defined using the ECASS II criteria [36] on a CT scan at approximately 24 h post therapy (VLF: hemorrhage type *p* = 0.63, occlusion side and hemorrhage × occlusion interaction *p* < 0.01; LF: hemorrhage type *p* = 0.73, occlusion side *p* < 0.01, and interaction *p* = 0.01). There were no significant differences in the gain (VLF: hemorrhage type *p* = 0.59, occlusion side *p* = 0.97, interaction *p* = 0.82; LF: hemorrhage type *p* = 0.84, occlusion side *p* = 0.70, interaction *p* = 0.76) or the phase (VLF: hemorrhage type *p* = 0.34, occlusion side *p* = 0.50, interaction *p* = 0.52; LF: hemorrhage type *p* = 0.95, occlusion side *p* = 0.15, interaction *p* = 0.76) in the <24 h after the LSW time period. There were no statistically significant differences in any of the transfer function parameters relative to the hemorrhage during any of the other time points.

The end-tidal CO_2_ levels were shown to demonstrate a temporal trend with higher levels in patients with incomplete recanalization (TICI 2b or less) compared with TICI 3. This was statistically significant after 96 h, *p* = 0.03 (Figure 3).

## 4. Discussion

This study examined the time-dependent impairments in cerebral autoregulation following an endovascular thrombectomy. We found that patients with complete (TICI 3) recanalization after the thrombectomy had less severe impairments in their autoregulation than patients with incomplete recanalization (TICI 2a or 2b). Impairments in autoregulation were primarily observed 24 to 96 h after the LSW time and were bilateral, affecting both the ipsilateral and contralateral hemispheres, supporting prior work demonstrating the presence of global impairments in autoregulation after an ischemic stroke [10,13,37]. However, some prior studies have demonstrated the presence of unilateral impairments in cerebral autoregulation [12,38,39], suggesting an element of interpatient variability, with some suggesting that large-vessel atherosclerosis and small-vessel disease likely contribute to bilateral impairments [37,39].

To our knowledge, this is the first study to examine the effects of the degree of recanalization post thrombectomy on cerebral autoregulation. These findings have important implications for patient management. First, the observation that incomplete recanalization may place patients at risk for less efficient autoregulation provides additional support for technique optimization aimed at achieving complete (TICI 3) recanalization, particularly given prior data showing that impairments in autoregulation correlate with larger infarct sizes and worse long-term outcomes [12,13]. Second, patients with incomplete revascularization may need closer hemodynamic monitoring and more tightly regulated blood pressure goals in the immediate post-stroke period lasting up to 96 h after the last seen well time. Even accounting for differences in the infarct volume between the two groups, cerebral autoregulation was significantly less effective in patients with incomplete recanalization. Given that autoregulatory impairments were most prominent between the 24 and 96 h post-LSW times, intensive blood pressure monitoring and control may be indicated up for up to 4 days post-LSW time in these patients. A limitation to this conclusion is the difficulty in determining the directionality of the effects between the degree of recanalization, the cerebral autoregulation, and the infarct volume. There may be specific characteristics of the vasculature that may predispose a subset of patients to poorer recanalization, worse cerebral autoregulation profiles, and larger infarct volumes.

The temporal changes in the end-tidal CO_2_ demonstrated in this study highlight the importance of monitoring this variable in large vessel occlusion ischemic stroke patients, especially in the immediate peri-procedural and post-procedural period, given its known effect on cerebral blood flow [18,19]. The higher end-tidal CO_2_ levels in patients with incomplete recanalization, especially at later time points, is intriguing and may reflect higher infarct volumes, which this variable (end-tidal CO_2_) was shown to be associated with. However, this finding is not sufficient to account for all the between-group differences noted in the cerebral autoregulation, which were demonstrated early, while the end-tidal CO_2_ differences were more obvious at time points beyond 96 h. 

Notably, the hemorrhagic transformation (PH1–PH2) [36] was primarily associated with early (<24 h post-LSW time) unilateral impairments in autoregulation, affecting the side with the LVO. This association between the early unilateral impairments in cerebral autoregulation and hemorrhagic transformation is consistent with prior work: in patients who did not receive an EVT, impairments in their autoregulation <6 h following the LSW time were shown to be associated with increased rates of hemorrhage at 24 h post-LSW time [40]. 

There are inherent limitations of the TCD for continuous blood flow monitoring. TCD provides a measure of the blood flow velocity but not the flow rate. However, the former is an adequate surrogate for the latter as long as the diameter of the insonated artery remains constant. Given that we acquired data in the semi-supine position and relied on spontaneous measurements, we did not anticipate any changes in the observed artery diameter. We also note that autoregulation in the examined vascular territories (bilateral MCA) may not be reflective of autoregulation in other vascular territories. Second, the requirement of adequate acoustic windows to obtain data and difficulties with identifying completely occluded versus partially occluded vessels in the absence of transcranial color-coded duplex sonography can lead to increases in interobserver variability. We attempted to minimize the interobserver variability by ensuring all our TCDs were performed by three professional sonographers (P.C., A.M., and S.L.). Third, we used conventional frequency ranges to assess cerebral autoregulation. However, as noted before [27,28], these were chosen to standardize the assessment of autoregulation across studies, and in reality, autoregulation is most effective at frequencies at or lower than 0.03 Hz [29]. Fourth, cerebral autoregulation may be affected by multiple variables that were not adequately assessed in our study, such as medications [41], pre-morbid vascular risk factors, and critical illness, including sepsis [42]. Exploration of the impact of these factors on the temporal course of cerebral autoregulation remains for future studies.

Despite its limitations, our results suggest that dynamic cerebral autoregulation was impaired in a time-dependent manner after an LVO stroke and an EVT, and complete recanalization (TICI 3) was associated with an improved autoregulatory profile compared to an EVT with incomplete recanalization. These results strongly suggest that patients with incomplete recanalization may require closer hemodynamic monitoring up to 4 days post thrombectomy. 

## Figures and Tables

**Figure 1 brainsci-10-00641-f001:**
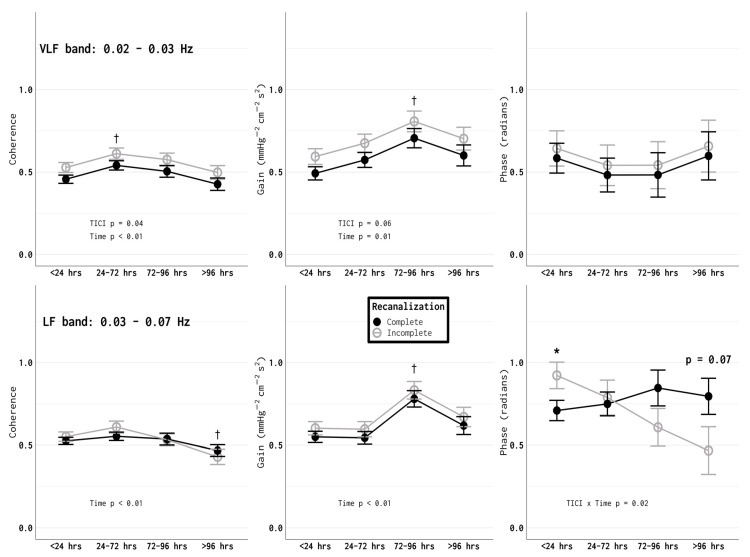
An improved TICI (Thrombolysis in Cerebral Infarction) scores in patients who received IAT (intra-arterial therapy) was associated with an improved autoregulation profile. In the VLF band, the coherence was higher in patients with an incomplete (TICI 2a or 2b) recanalization (*p* = 0.04). In the LF band, there was a similar trend in the coherence, but this difference did not achieve statistical significance (*p* = 0.07). The gain tended to be higher in patients with incomplete recanalization in the VLF band, but this difference did not reach statistical significance (*p* = 0.06). For both the incomplete and complete recanalization groups, the gain peaked at the 72–96 h time point (*p* = 0.01 for the VLF band, *p* < 0.01 for the LF band). No clear trends in the phase were observed in the VLF band. In the LF band, the phase increased over time in patients with complete recanalization and decreased over time in patients with incomplete recanalization (*p* = 0.02). * *p* < 0.05, incomplete vs. complete recanalization, † *p* < 0.05 vs. <24 h time point.

**Figure 2 brainsci-10-00641-f002:**
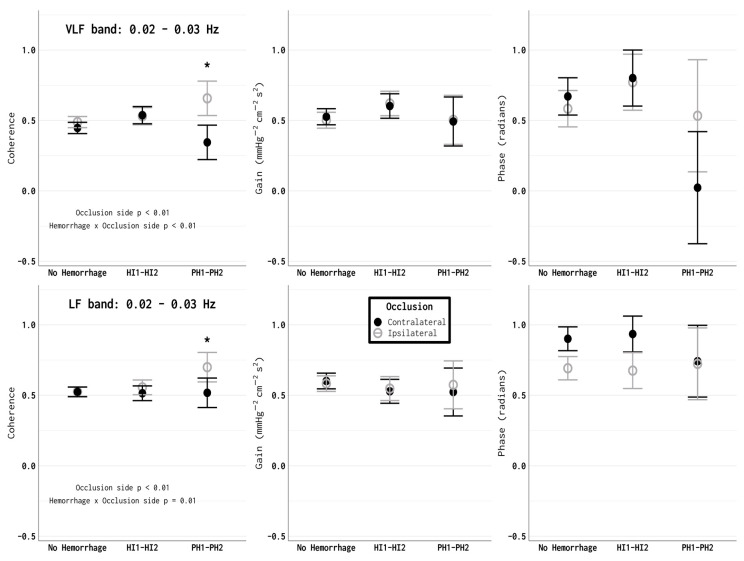
A PH1–PH2 (parenchymal hematoma types 1 and 2) hemorrhagic transformation was associated with impaired autoregulation in the ipsilateral hemisphere. In both of the VLF and LF bands, coherence on the ipsilateral side was significantly higher compared to the contralateral hemisphere in patients with a PH1–PH2 hemorrhagic transformation (* *p* < 0.01). The phase and gain were not significantly different between the ipsilateral and contralateral hemispheres in this group (for both the VLF and LF bands).

**Figure 3 brainsci-10-00641-f003:**
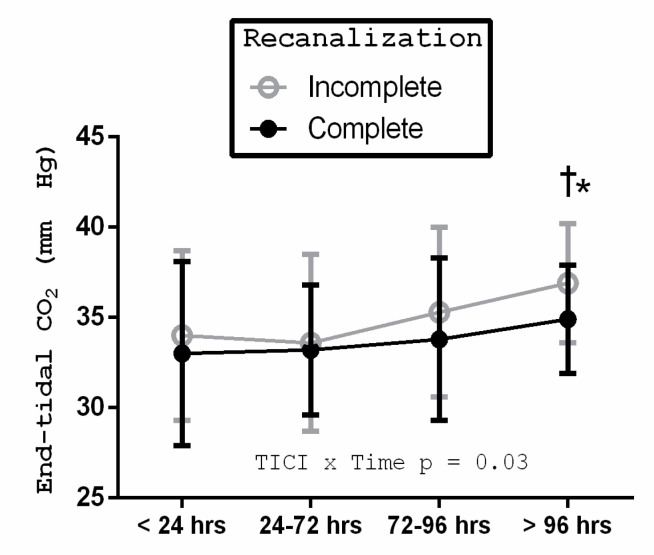
End-tidal CO_2_ trends with time. The end-tidal CO_2_ was decreased in the <24 h compared to the late time point of the study (time *p* < 0.01). After 96 h, incomplete recanalization was significantly associated with a higher end-tidal CO_2_ (* *p* < 0.01). The mean and SD values allowed for discrimination.

**Table 1 brainsci-10-00641-t001:** Subject demographics and clinical data.

	Demographic/ Clinical Variable	TICI 2a/2b (*n* = 39)	TICI 3 (*n* = 68)	*p*-Value ^†^
Age, mean ± SD		68.6 ± 13.4	70.2 ± 12.8	0.76
Male sex		22 (56.4%)	36 (52.9%)	0.34
Received tPA		15 (38.5%)	30 (44.1%)	0.57
LSW to tPA time (min), mean ± SD		156.5 ± 63.7	185.4 ± 93.4	0.34
LSW to EVT time (min), mean ± SD		424.7 ± 286.5	456.0 ± 230.5	0.35
LVO location	L MCA	15 (38.4%) M1: 15, M2: 0	25 (36.8%) M1: 19, M2: 6	0.67
	R MCA	20 (51.3%) M1: 18, M2: 2	34 (50%) M1: 29, M2: 5	
	L ICA	1 (2.6%)	6 (8.8%)	
	R ICA	3 (7.7%)	3 (4.4%)	
ASPECTS score, median (IQR)		9.0 (7–10)	9.0 (7–10)	0.89
Collateral grading, median (IQR)		2.0 (2–3)	2.0 (2–3)	0.56
NIHSS on admission, median (IQR)		13.0 (9–16)	15.0 (11–18)	0.45
NIHSS on discharge, median (IQR)		7.0 (2–17)	5.0 (2–11)	0.58
Early neurological recovery *		21 (53.8%)	51 (73.9%)	**0.03**
MRS at 30 days, median (IQR)		3.0 (2–5)	3.0 (2–4)	0.87
MRS at 90 days, median (IQR)		2.0 (2–4)	3.0 (2–4)	0.43
Infarct volume (mL), mean ± SD		74.5 ± 89.9	34.3 ± 63.6	**<0.01**
Significant hemorrhage		6 (15.4%)	6 (8.8%)	0.30
Midline shift		1 (10.3%)	1 (7.3%)	0.67
LSW to TCD time, <24h, mean ± SD		13.9 ± 3.3	13.9 ± 5.2	0.78
LSW to TCD time, 24–72 h, mean ± SD		37.9 ± 9.5	44.0 ± 13.5	0.13
LSW to TCD time, 72–96 h, mean ± SD		85.6 ± 6.5	86.6 ± 7.3	0.46
LSW to TCD time, 96 h +, mean ± SD		155.5 ± 36.0	120.8 ± 19.8	**<0.01**
End-tidal CO_2_, <24 h, mean ± SD		34.0 ± 4.7	33.0 ± 5.1	**0.01** ^‡^
End-tidal CO_2_, 24–72 h, mean ± SD		33.6 ± 4.9	33.2 ± 3.6	
End-tidal CO_2_, 72–96 h, mean ± SD		35.3 ± 4.7	33.8 ± 4.5	
End-tidal CO_2_, >96 h, mean ± SD		36.9 ± 3.3	34.9 ± 3.0	

Values are presented as mean ± SD, median (IQR), or *n* (%), where appropriate. Boldface represents statistically significant *p*-values. * Defined as an NIHSS (National Institute of Health Stroke Score) score of 0–2 at discharge and/or improvement in the NIHSS score at discharge of ≥ 4. ^†^ The *p*-value for differences between groups of a *t*-Test, a Mann–Whitney test, or a chi-square test, except in ^‡^, where a repeated measures ANOVA was used. ASPECTS: Alberta Stroke Program Early CT Score, EVT: endovascular thrombectomy, ICA: internal carotid artery, LSW: last seen well, LVO: large vessel occlusion, MCA: Middle Cerebral Artery, mRS: modified Rankin Score, TCD: Transcranial Doppler, TICI: Thrombolysis in Cerebral Infarction score, tPA: tissue Plasminogen Activator.

**Table 2 brainsci-10-00641-t002:** Infarct size vs. cerebral autoregulation parameters, as a mean (SD), across all time points.

VLF (0.02–0.03 Hz)			
Parameter of autoregulation	Stroke Volume < 70 mL	Stroke Volume > 70 mL	*p*-Value
Ipsilateral coherence	0.497 (0.251)	0.606 (0.219)	**0.027**
Contralateral coherence	0.473 (0.236)	0.532 (0.256)	0.221
Ipsilateral gain	0.602 (0.394)	0.495 (0.281)	0.147
Contralateral gain	0.644 (0.400)	0.625 (0.415)	0.81
Ipsilateral phase	0.620 (0.879)	0.263 (0.578)	**0.03**
Contralateral phase	0.599 (0.935)	0.540 (0.701)	0.742
**LF (0.03–0.07 Hz)**			
**Parameter of autoregulation**	**Stroke Volume < 70mL**	**Stroke Volume > 70 mL**	***p*-Value**
Ipsilateral coherence	0.537 (0.197)	0.582 (0.207)	0.254
Contralateral coherence	0.517 (0.193)	0.489 (0.199)	0.468
Ipsilateral gain	0.630 (0.357)	0.475 (0.189)	**0.018**
Contralateral gain	0.660 (0.324)	0.579 (0.373)	0.229
Ipsilateral phase	0.783 (0.502)	0.338 (0.499)	**<0.001**
Contralateral phase	0.912 (0.508)	0.561 (0.606)	**0.001**

Boldface represents statistically significant *p*-values. VLF—very low frequency, LF—low frequency.

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
