# Peer review of "Dynamic Cerebral Autoregulation Post Endovascular Thrombectomy in Acute Ischemic Stroke"

_brainsci, 2020, doi:10.3390/brainsci10090641_

Round 1

Reviewer 1 Report

Authors havee written an interesting and important article about monitoring cerebral autoregulation. Few things have to be clarified however.

Materials and Methods:

p.2, line 83: Please provide more detailed information or a reference for TICI -scores.

p.3, line 103: Please write out abbreviations "VLF" and "LF".

p.3, line 139: Please write out abbreviation "PVCs".

Results:

p.3-4, lines 142-144: Did patients use any medications (for example calcium channel blockers)? Were there any differences in medications between groups? Please provide information about this. Add also a short mention about drugs in discussion section (for example to p.8, line 253 if drugs might be one confounding factor). 

Tables and Figures:

Please write out all abbreviations in figure legends/under tables.

There is no mention about Figure 2 in main text. Maybe it could be mentioned for example on p.7 line 195 "Impairment in cerebral autoregulation within the first 24 hours after stroke onset was also associated with hemorrhagic transformation (Figure 2)." 

Author Response

Reviewer 1

Comments and Suggestions for Authors

Authors have written an interesting and important article about monitoring cerebral autoregulation. Few things have to be clarified however.

Response: We appreciate the reviewer’s thoughtful review and helpful comments.

Materials and Methods:

p.2, line 83: Please provide more detailed information or a reference for TICI -scores

Response: Reference from original paper by Higashida inserted (see line 90; reference 20).

p.3, line 103: Please write out abbreviations "VLF" and "LF".

Response: Long forms provided in text – now on line 111

p.3, line 139: Please write out abbreviation "PVCs".

Response: Long forms provided in text – now on line 149/150

Results:

p.3-4, lines 142-144: Did patients use any medications (for example calcium channel blockers)? Were there any differences in medications between groups? Please provide information about this. Add also a short mention about drugs in discussion section (for example to p.8, line 253 if drugs might be one confounding factor). 

Response: The reviewer makes an important comment – we collected data prospectively on medications administered however given the wide variability in management of patients between the two institutions and within providers at the same institution including medication dose variability, this data could not be quantitatively analyzed in a meaningful fashion. We have stated this as a limitation in our discussion section with an associated reference– please see line 295; reference 41 .

Tables and Figures:

Please write out all abbreviations in figure legends/under tables.

Response: Done.

There is no mention about Figure 2 in main text. Maybe it could be mentioned for example on p.7 line 195 "Impairment in cerebral autoregulation within the first 24 hours after stroke onset was also associated with hemorrhagic transformation (Figure 2)." 

Response: Done  (Line 222)

Reviewer 2 Report

This manuscript provides a much needed data on autoregulatory status of individuals with large vessel occlusion (LVO) undergoing endovascular therapy (EVT). 107 patients were included and helpfully, radiological diagnosis of LVO was made. Crucially, the authors have interpreted the haemodynamic findings in the context of identification of those who may warrant greater monitoring - a concept that has been reinforced by large BP focussed studies (which have been referenced in the introduction).

Introduction:

There needs to be a greater emphasis on the peri-procedural impact i.e. anaesthesia, pre-EVT BP control and post-procedural monitoring. I suggest review and referencing PMID: 31813569. Specifically as there needs to be acknowledgement for potential carbon dioxide changes (which have not necessarily been addressed enough in this analysis despite good confirmative data existing).

Methods: 

30 degree head of bed position - needs a reference for justification

29 patients did not have useable data - explain? This is 21% - higher than explained by window issues - we need to understand some of the difficulties associated with data collection in this context in this novel study.

Results:

There needs to be some greater analysis and interpretation for potential EtCO2 differences in those who recanalised full (TICI 3) as compared to partial (TICI 2a or 2b). TICI 3 appeared to be consistently hypocapnic thoughout (and to a greater degree) - this needs assessment based on infarct size and recanalisation status to determine significant differences. This is crucial as we know ischaemic stroke patients are hypocapnic (PMID: 31102829), though this meta-analysis lacked patients undergoing EVT as a sub-group - this provides very helpful further data to reinforce this area of work.

Table 1 - needs comparisons between groups to highlight statistically significant differences (for example - did those with TICI 3 receive significantly more tPA?).

General vs concious sedation data should be included in Table 1.

Given the strong relationship between CO2 and autoregulatory parameters - it would be pertinent to see trend of CO2 with Figure 1 to demonstrate any temporal associations exerting an influence?

Figure 2 - not particularly clear or visible - quality and layout needs revision please (suggest a clearer representation - also darker should be ipsilateral perhaps and lighter contralateral)

Overall - very appropriate conclusions drawn and a novel and important dataset - however, given the importance of the peri-operative period and CO2 change - we need some attention to this throughout.

Author Response

Comments and Suggestions for Authors

This manuscript provides a much needed data on autoregulatory status of individuals with large vessel occlusion (LVO) undergoing endovascular therapy (EVT). 107 patients were included and helpfully, radiological diagnosis of LVO was made. Crucially, the authors have interpreted the haemodynamic findings in the context of identification of those who may warrant greater monitoring - a concept that has been reinforced by large BP focused studies (which have been referenced in the introduction).

Response: We appreciate the reviewer’s thoughtful review and helpful comments.

Introduction:

There needs to be a greater emphasis on the peri-procedural impact i.e. anaesthesia, pre-EVT BP control and post-procedural monitoring. I suggest review and referencing PMID: 31813569. Specifically as there needs to be acknowledgement for potential carbon dioxide changes (which have not necessarily been addressed enough in this analysis despite good confirmative data existing)

Response: Please see lines 64-68 where these important observations are now addressed and the appropriates references now inserted (references 18 and 19)

Methods: 

30 degree head of bed position - needs a reference for justification

Response: Please see lines 104-105  where an appropriate reference has been inserted(reference 25)

29 patients did not have useable data - explain? This is 21% - higher than explained by window issues - we need to understand some of the difficulties associated with data collection in this context in this novel study.

Response: This astute observation has been addressed in the Results section. Please see lines 150 to 152.

Results:

There needs to be some greater analysis and interpretation for potential EtCO2 differences in those who recanalised full (TICI 3) as compared to partial (TICI 2a or 2b). TICI 3 appeared to be consistently hypocapnic thoughout (and to a greater degree) - this needs assessment based on infarct size and recanalisation status to determine significant differences. This is crucial as we know ischaemic stroke patients are hypocapnic (PMID: 31102829), though this meta-analysis lacked patients undergoing EVT as a sub-group - this provides very helpful further data to reinforce this area of work.

Response: We completely agree with the reviewer; this has now been addressed in the Results and Discussion sections. We have also stated this as a limitation given the difficulty in adequately measuring EtCO2 level in non-intubated patients.

Table 1 - needs comparisons between groups to highlight statistically significant differences (for example - did those with TICI 3 receive significantly more tPA?).

Response: Please see table 1; this has been duly addressed. Of note, there was no difference between tPA administration between the two subgroups. Only a few variables had a statistically significant difference between the two subgroups including infarct volume which was lower in the TICI 3 group as expected and early neurologic recovery which was higher in this group;  last seen well (LSW) to TCD in the 96 hour subgroup which was shorter in the TICI 3 group likely reflective of shorter ICU/hospital stays necessitating earlier studies although we did not specifically look at ICU length of stay. Finally there was a time dependent difference in end tidal C02 which became statistically significant at 96 hours which is further discussed under subsequent comments by the author.

General vs conscious sedation data should be included in Table 1.

Response: This is an important factor; unfortunately this data was not prospectively collected.

Given the strong relationship between CO2 and autoregulatory parameters - it would be pertinent to see trend of CO2 with Figure 1 to demonstrate any temporal associations exerting an influence?

Response: This is an important observation; we therefore included figure 3 which demonstrates a temporal trend of increasing end-tidal C02 with time and a significant difference at time point > 96 hours between the TICI 3 compared with TICI 2b or less. See lines 234 to 240. Given missing data especially in patients who were not intubated, it was not possible to do a thorough analysis of the association between cerebral autoregulation parameters and ETCO2 levels. Additional paragraphs in the Discussion section (lines 268 to 274, 297 to 299)  were also added.

Figure 2 - not particularly clear or visible - quality and layout needs revision please (suggest a clearer representation - also darker should be ipsilateral perhaps and lighter contralateral)

Response: A higher clarity figure is now attached (see Figure 2.tiff)  

Overall - very appropriate conclusions drawn and a novel and important dataset - however, given the importance of the peri-operative period and CO2 change - we need some attention to this throughout.

Response: We appreciate this reviewer’s in-depth and highly critical review of our work; the content of the manuscript has been substantially improved as a result.

Round 2

Reviewer 2 Report

I would like to thank the authors for kindly addressing my comments and suggestions in full. The additional analysis on temporal EtCO2 trends post EVT is fascinating and should propogate further work in this rapidly evolving area.